# Thermal Effects in Dissimilar Magnetic Pulse Welding

**Joerg Bellmann** [1,2,*] **, Joern Lueg-Althoff** [3] **, Sebastian Schulze** [2] **, Marlon Hahn** [3] **,
Soeren Gies** [3] **, Eckhard Beyer** [1,2] **and A. Erman Tekkaya** [3]

1   Institute of Manufacturing Science and Engineering, Technische Universitaet Dresden, George-Baehr-Str. 3c,
    01062 Dresden, Germany; eckhard.beyer@tu-dresden.de
2   Business Unit Joining, Fraunhofer IWS Dresden, Winterbergstr. 28, 01277 Dresden, Germany;
    sebastian.schulze@iws.fraunhofer.de
3   Institute of Forming Technology and Lightweight Components, TU Dortmund University, Baroper Str. 303,
    44227 Dortmund, Germany; joern.lueg-althoff@iul.tu-dortmund.de (J.L.-A.);
    marlon.hahn@iul.tu-dortmund.de (M.H.); soeren.gies@iul.tu-dortmund.de (S.G.);
    erman.tekkaya@iul.tu-dortmund.de (A.E.T.)
*   Correspondence: joerg.bellmann@tu-dresden.de; Tel.: +49-351-83391-3716

**Abstract:** Magnetic pulse welding (MPW) is often categorized as a cold welding technology,
whereas latest studies evidence melted and rapidly cooled regions within the joining interface.
These phenomena already occur at very low impact velocities, when the heat input due to plastic
deformation is comparatively low and where jetting in the kind of a distinct material flow is not
initiated. As another heat source, this study investigates the cloud of particles (CoP), which is
ejected as a result of the high speed impact. MPW experiments with different collision conditions
are carried out in vacuum to suppress the interaction with the surrounding air for an improved
process monitoring. Long time exposures and flash measurements indicate a higher temperature in
the joining gap for smaller collision angles. Furthermore, the CoP becomes a finely dispersed metal
vapor because of the higher degree of compression and the increased temperature. These conditions
are beneficial for the surface activation of both joining partners. A numerical temperature model
based on the theory of liquid state bonding is developed and considers the heating due to the CoP
as well as the enthalpy of fusion and crystallization, respectively. The time offset between the heat
input and the contact is identified as an important factor for a successful weld formation. Low values
are beneficial to ensure high surface temperatures at the time of contact, which corresponds to the
experimental results at small collision angles.

**Keywords:** magnetic pulse welding; dissimilar metal welding; solid state welding; welding window;
cloud of particles; jet; surface activation

## 1. Introduction

Dissimilar metal welding plays an important role in the fabrication of multi-material parts.
Materials with different mechanical, physical, or chemical properties need to be joined in order to fulfill
the requirements of lightweight structures and high endurance parts or to save costs. Conventional
thermal joining technologies reach their limits because of the formation of brittle intermetallic phases
during the welding process. Decreasing the heat input is expedient to generate sound welds and to
sustain the properties of the base materials. Impact welding processes like explosive welding (EXW)
and magnetic pulse welding (MPW) utilize the oblique high-speed collision between two metallic
surfaces to achieve metallurgical connections [1]. MPW is a suitable joining technology for dissimilar

metal welding of tubular parts such as hybrid drive shafts [2]. In the initial state, the two joining partners (a movable "flyer" and a stationary "parent") are positioned with some standoff $g$, which defines the acceleration distance. During EXW, the acceleration of the flyer part is achieved by the detonation of an explosive, which is applied on the flyer. In contrast to that, the flyer acceleration in MPW is driven by a magnetic pressure. This pressure is the result of a sudden discharge of a capacitor bank via a tool coil that is positioned in close vicinity to the flyer. If the flyer material is electrically conductive, opposing eddy currents are induced into the flyer and resulting Lorentz forces drive the flyer away from the tool coil. A surface-related mathematical equivalent of the volume Lorentz forces is the magnetic pressure, which also causes shock loads on the tool coils and limits their lifetime. Comprehensive reviews of EXW and MPW can be found in [3,4], respectively. The weld is formed while both surfaces are pressed together during the collision. Interface pressures, in the order of several GPa, occur for a duration of a few microseconds. The microstructures of explosive welds show grains at the interface that underwent large plastic deformations. Strain rates can reach the order of $10^4$ to $10^5$ s$^{-1}$ and, consequently, the material behaves like a fluid. The "jetting" effect is a consequence of the hydrodynamic phenomena taking place at the propagating collision point. The jet is often described as a massive material flow and is supposed to clean and activate the surfaces before welding. Within the last decades, the influence of thermal aspects during solid state welding was brought into the focus of many researchers, aiming for the identification of the relevant joining mechanism(s). For example, Ishutkin et al. found for EXW that the temperature of the shock compressed gas in the joining gap reaches several thousand °C and causes surface melting of both joining partners [5]. Together with the prevalent pressure, the overall energy input might be beyond the upper welding boundary and result in bad weld quality due to heat accumulation at the interface. This phenomenon occurs especially at positions that are exposed to the extensive heating for a longer time, i.e. further behind the initial collision point. Additionally, the temperature in the joining gap is strongly influenced by the thermodynamic properties of the medium in the joining gap. Deribas and Zakharenko describe the formation of a "cloud of particles" (CoP) that contributes to the temperature increase [6]. Thus, special interlayers are applied during EXW to avoid excessive intermetallic phase formation, see [7]. Another possibility is to reduce the "levels of temporal and force parameters required for joint formation" if the temperature of the near-contact layer of the welded materials is too high. This suggestion was made by Lysak and Kuzmin according to their pressure-temperature-time model, described in [8]. The latest descriptions of EXW and MPW weld seams have several aspects in common: melted and rapidly cooled metal layers were observed for EXW by Bataev et al. [9] and for MPW by Stern et al. [10]. Sharafiev et al. reported sharp boundaries between the base materials and an intermediate layer of impact welded A-Al joints. New, recrystallized grains in the size of nanometers and high dislocation densities give evidence for melting and rapid solidification [11]. Amorphous structures have also been described by metallurgists [12,13] that can be attributed to solidification with cooling rates in the order of $10^7$ K/s [9]. Thus, it can be assumed that "liquid state bonding" is an occurring joining mechanism for EXW as well as MPW [14]. Nevertheless, different theories exist about the most relevant source of the thermal energy. In [9,15], the thermal energy is related to the plastic deformation during the collision, while in [5,16] it is attributed to the shock compressed gas in the joining gap as well. Successful MPW experiments under vacuum conditions revealed that no initial gas is required in the joining gap. Nevertheless, this does not contradict the theory of a compressed cloud of particles in the joining gap. In MPW experiments, it was found that the impact pressure or the normal impact velocity can be reduced as far as the collision angle is small enough [17]. Thus, less energy is needed for the flyer acceleration and the life time of the tool coils is increased. Furthermore, additional energy from an exothermic reaction between the joining partners or interlayers is beneficial for the joint formation, if well-adjusted [18]. This underlines the influence of thermal effects in MPW and the need for a comprehensive model or at least the identification of the most relevant input variables for heating and cooling of the surfaces and the interface. There are already thermal models existing for MPW [9,13,15,19] but they do not take the CoP into consideration as a heat source. The time

for solidification has also been calculated in previous publications [9,20,21] but without considering dissimilar materials, the temperature distribution after the heating and not to mention the effect of phase transformations. For the prediction of the weld formation it is important to know the time dependent temperature in the joining zone. The weld formation might be hindered if the surface temperatures are too low at the time of contact or bounce back effects occur [21] before the solidification of the welding interface is completed.

The objectives of this experimental and numerical work can be summarized as follows:

1. Investigate the influence of the flyer kinetics on the material flow.
2. Study the influence of different collision conditions on the formation and properties of the jet or "cloud of particles" (CoP) and the corresponding thermal conditions in the joining gap.
3. Build up a temperature model for the welding interface, based on the heat input by the CoP.

## 2. Materials and Methods

### 2.1. Nomenclature

Three different experimental setups and a numerical model with a multitude of parameters will be introduced in the following chapters. Table 1 lists all symbols that are used within this paper in order to shorten the captions of figures and tables.

**Table 1.** Nomenclature for experimental and numerical setup.

| Symbol | Parameter | Symbol | Parameter |
|---|---|---|---|
| $A$ | Area | $s$ | Thickness of the flyer tube |
| $b$ | Equivalent thickness of the molten layer | S | High voltage switch |
| $c$ | Heat capacity | $t$ | Time |
| C | Capacitance; Contact point | $T$ | Temperature |
| $d$ | Distance to the impact location | $t_{con}$ | Contact time |
| $E$ | Charging energy | $t_{f,start}$ | Flash appearance time |
| $f_{discharge}$ | Discharge frequency | $T_{Fly}$ | Flyer temperature |
| $g$ | Initial joining gap | $T_{fus}$ | Melting temperature |
| $I$ | Discharge current | $t_{heat}$ | Heating time |
| $I_f$ | Intensity of the impact flash | $t_{imp}$ | Impact time |
| $I_{max}$ | Maximum discharge current | $T_{Par}$ | Parent temperature |
| $k$ | Thermal conductivity | $T_{vap}$ | Boiling temperature |
| $l$ | Length of welded zone | $t_{wait}$ | Waiting time |
| $l_c$ | Collision length | $U$ | Voltage |
| $L_i$ | Inner inductance of the pulse generator | $V$ | Volume |
| $l_w$ | Working length | $v_i$ | Impact velocity |
| $m$ | Mass | $v_{i,r}$ | Radial impact velocity |
| $p$ | Surrounding pressure | $w_c$ | Width of the coil concentration zone |
| $P$ | Heat input | $z$ | Distance perpendicular to the steel surface |
| $p_m$ | Magnetic pressure | $\alpha$ | Angle of inclined parent surface |
| $Q$ | Total heat input | $\gamma$ | Damping coefficient of $I(t)$ |
| $Q_s$ | Heat input to each surface | $\Delta H_{fus}$ | Enthalpy of fusion |
| Ra | Mean roughness index | $\Delta t$ | Gap closing time |
| $R_i$ | Inner resistance of the pulse generator | $\rho$ | Density |

### 2.2. Experiments

The material combination aluminum-steel was chosen for this study due to its relevance for current and future lightweight concepts in the transportation sector. For a good comparability with previous studies, the outer tube consists of the aluminum alloy EN AW-6060, while the inner rod is made of C45 [17,22]. The chemical compositions of the alloys are given in Table 2. Both parts were cleaned in ethanol before the joining experiments to remove debris from their surfaces.

**Table 2.** Aluminum EN AW-6060 alloy composition adapted from [23] and steel C45 (1.0503) alloy composition adapted from [24].

| Flyer Part EN AW-6060 [1], Quasi-Static Yield Strength Approximately 60 MPa [2] | | Parent Part C45 (1.0503), Normalized, Quasi-Static Yield Strength Approximately 490 MPa [3], Surface Polished (Ra = 1) | |
|---|---|---|---|
| Element | Weight % | Element | Weight % |
| Mg | 0.35–0.6 | C | 0.42–0.5 |
| Mn | ≤0.1 | Mn | 0.5–0.8 |
| Fe | 0.1–0.3 | P | <0.045 |
| Si | 0.3–0.6 | S | <0.045 |
| Cu | ≤0.1 | Si | <0.4 |
| Zn | ≤0.15 | Ni | <0.4 |
| Cr | ≤0.05 | Cr | <0.4 |
| Ti | ≤0.1 | Mo | <0.1 |

[1] T66 heat treated: one hour at 500 °C and naturally aged, [2] determined by tube tensile test, [3] adapted from [25].

In the basic setup, MPW experiments with different charging energies were performed in order to identify the minimum energy required for a continuous weld seam along the circumference. Therefore, the setup depicted in Figure 1 was connected to two different pulse generators, resulting in two different resonant circuits with their characteristic values listed in Table 3. The current $I(t)$ was measured for each trial using a Rogowski current probe CWT 3000 B from Power Electronic Measurements Ltd. (Nottingham, UK) and the maximum current amplitude $I_{max}$ was evaluated. The pulse generator MPW 50/25 (Bmax, Toulouse, France) was connected to a single turn working coil, while the magnetic pressure at the EmGen setup was generated by a five-turn coil with a field shaper. During the high speed collision between the flyer and the parent part, a characteristic flash occurs which is called impact flash [26]. The time-dependent course of the light emission was measured with the flash measurement system described previously [22]. It was triggered by the current signal of the generator, where the rise of the current was defined as $t = 0$ µs, see schematic oscilloscope in Figure 1. Thus, the starting time of the flash $t_{f,start}$, its duration and maximum intensity were analyzed. The welding result was checked with a manual peel test at four positions at the circumference: 0°, 90°, 180° and 270°. The position at the slot of the coil (0°) is of special interest during process adjustment due to the reduced magnetic field intensity. However, the influence of the slot is limited to approximately 10% of the circumference. Hence, the metallographic analysis of the welding interface was performed at the opposed position (180°), since it is representative for almost the complete circumference [27]. Selected welding trials were performed with anodized flyer tubes (layer thickness 5 µm), which enable the reconstruction of the material flow [28].

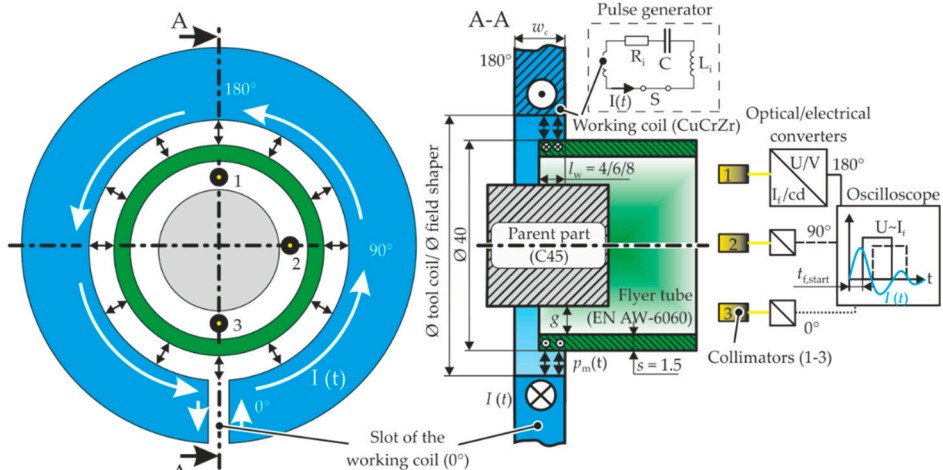

**Figure 1.** Basic setup for magnetic pulse welding (MPW) of tubes to cylinders (all values in millimeters, not true to scale, position of all parts is fixed during MPW).

**Table 3.** Characteristics of the resonant circuits and the deployed pulse generators.

| Setup | Unit | Bmax MPW 50/25 | EmGen |
|---|---|---|---|
| Capacitance | μF | 160 | 140 |
| Inductance [1] | nH | 372 | 2700 |
| Maximum charging energy | kJ | 32 | 40 |
| Maximum charging voltage | kV | 20 | 24 |
| Applied charging energy—$E$ | kJ | 4.5–9.6 | 7.0–22.7 |
| Discharge frequency [1]—$f_{discharge}$ | kHz | ~21 | ~9 |
| Damping coefficient $\gamma$ [1] | 1/s | 16,500 | 2700 |

[1] for the complete resonant circuit with working coil, field shaper and workpieces.

The flyer kinetics at the lower process boundary were studied for both pulse generators with the modified setup shown in Figure 2 while the parameter $d$ and, thus, the initial collision point was increased stepwise from zero to five millimeters from the flyer edge. After the flyer was sheared from the parent part, the length and location of the weld seam were measured at the 90°, 180° and 270° position for each test.

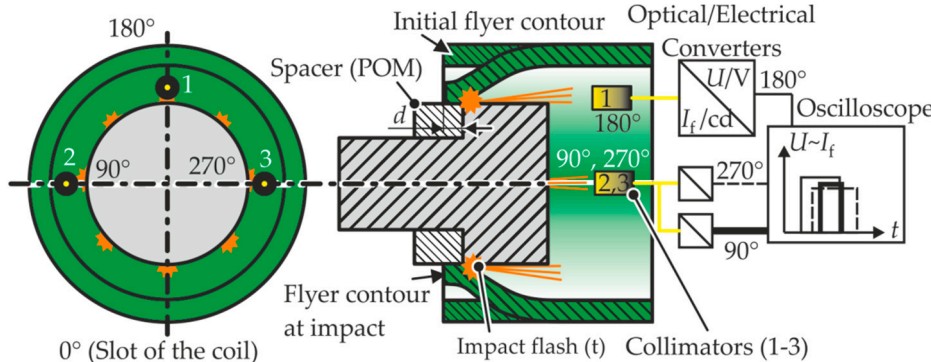

**Figure 2.** Modified MPW setup for investigation of the flyer kinetics with collimators (1–3) for flash detection (not true to scale, position of all parts is fixed during MPW).

For the third experimental part, the MPW process was carried out in a vacuum chamber as depicted in Figure 3, using the Bmax pulse generator. The surrounding pressure $p$, the collision length $l_c$, the working length $l_w$ as well as the contour of the parent part were varied. The camera Canon EOS 700D with an exposure time of six seconds, a fixed aperture of F13 and the light intensity ISO 100 was placed behind a translucent Plexiglas disc to take longtime exposures and to identify the color of the process glare. The average R/G/B value of 5x5 pixels was converted to the 2-D xy-color chart. This enables the estimation of the temperature in the joining gap, if an ideal black body emission is assumed. Again, the flash measurement system was used to detect the temporal course of the light intensity. Furthermore, a translucent plastic disc was placed inside of the flyer tube to study the interaction with the jet or cloud of particles, respectively. Qualitative conclusions can be drawn from the location where the debris sticks on the disc or if the plastic sheet is fractured. Additionally, the locations of the jet residues at the inner flyer surfaces were analyzed.

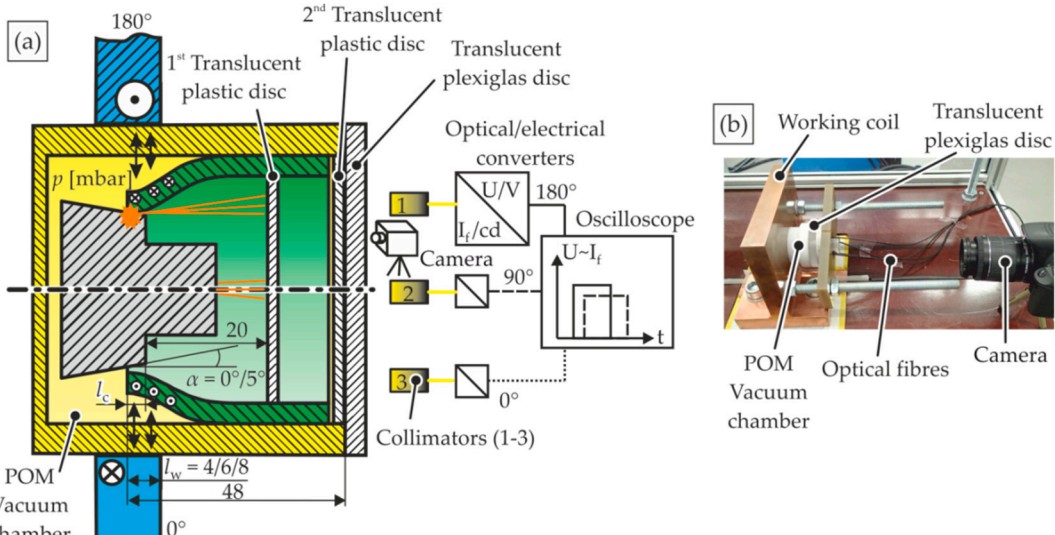

**Figure 3.** Modified setup for MPW at different surrounding pressures *p* with collimators (1–3) for flash detection and camera (**a**) schematic (not true to scale) and (**b**) photograph (position of all parts is fixed during MPW).

## 2.3. Numerical Simulations

The aim of the numerical investigations is to estimate the heating and cooling of the surfaces of the joining partners and predict the weldability based on the liquid state bonding theory. Experiments will be described in the following chapters, which reveal temperatures in the closing joining gap that are far beyond the fusion temperatures of both joining partners. The CoP is assumed to be a main heat source before the contact and, thus, the model should answer the question, whether liquid state bonding can occur under these conditions. Although the temperature of the CoP is experimentally estimated in this paper, its heat transfer to the surfaces is very hard to determine. The density or mass, respectively, and the surface coefficient for heat transfer are difficult to access and, thus, the strategy performed in [15,29,30] is applied: The amount of melted flyer and parent material is quantified in polished cross sections and serves as an upper boundary of the heat input. It is cross-checked with the kinetic flyer energy, which determines the overall limit of the introduced energy. Furthermore, the following assumptions were made for the model:

1.  The thermal energy of the CoP is responsible for the surface activation before both joining partners get into contact. In order to simplify the numerical model, this is assumed to be the only heat source. The heat input by plastic deformation after the collision is not considered in the model.
2.  The thermal energy of the CoP is equally distributed to both joining partner surfaces, which seems admissible for small collision angles.
3.  At the welding interface, just solid and liquid phases are present at the time of contact. If the surface temperature would lead to vaporization before contact, the material in the gaseous phase would be pressed out of the joining gap together with the CoP during MPW, provided a sufficient collision angle.
4.  The influence of the temperature on the materials' densities, heat capacities and thermal conductivities is not considered in the simulations.

The one-dimensional numerical model for the calculation of the time-dependent surface temperatures was set up within the commercial software COMSOL Multiphysics® (Version 5.2, COMSOL Multiphysics GmbH, Goettingen, Germany). The model was simplified compared to the real MPW process that consists of three stages: The initial collision depicted in Figure 4a, the CoP formation, shown in Figure 4b and the movement of the collision point C, see Figure 4c. During the simulation, the flyer and parent parts were fixed with a constant gap of 2 µm, as shown in Figure 4d.

The CoP formation was not implemented in the model, but its heat input to both joining partners as well as the heat losses by conduction in the parts were taken into account. In order to recreate the moving contact point C during the real MPW process with the fixed joining partners, the following strategy was applied in the numerical model: During the surface activation by the CoP in Figure 4b, any heat transfer through the gap was suppressed. But then, at the contact time $t_{con}$ in the real MPW process, the heat conductivity of the gap was set to an extremely high value of $10^{10}$ W/(mK) and, thus, the intimate contact between both joining partners as shown in Figure 4c was imitated without the need for moving parts or meshes within the simulation. The implemented temporal course of the heat input $P$ and the contact time $t_{con}$ are depicted in Figure 4e. The following time steps were chosen for the numerical simulations: 0.02 µs for 0 µs $< t <$ 10 µs and 0.5 µs for 10.5 µs $< t <$ 100 µs. In order to study the temperature distribution in the close vicinity to the welding interface, the element size was set to 0.1 µm. Due to the one dimensional character of the model, the heat sources were defined with a certain area density at two points on both surfaces, see Figure 4d. During the simulations, different heat quantities $Q_S$, heating times $t_{heat}$ and material combinations were investigated. The relevant material parameters are listed in Table 4. The heating time at a certain point C strongly depends on the velocity of the CoP and the distance to the initial point of impact. Due to the accumulation of the CoP during the weld front propagation, the heat input and heating duration vary, too. Furthermore, a waiting time $t_{wait}$ was defined between the end of the heat input and the contact time $t_{con}$. By setting $t_{wait}$ to 0 µs, the heat input generated by the CoP is transferred to the plates in the vicinity of the contact line. This corresponds to the case when the CoP and the collision point C have the same velocity. If the CoP travels faster than the collision point, the waiting time is increased in the simulation. This procedure abstracts the effects in the moving interaction zone of a real MPW process. Nevertheless, it simplifies the modeling and reveals the most relevant influencing factors. Of course, these parameters have to be transferred to the kinematic process variables like collision front velocity or collision angle in a further step.

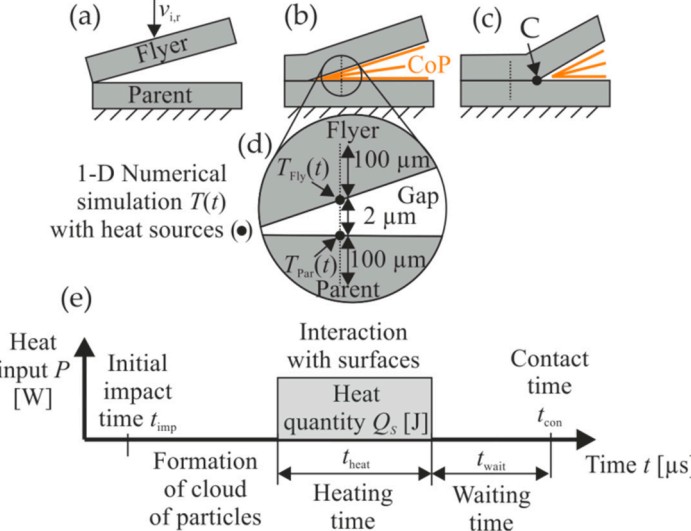

**Figure 4.** Process steps during MPW showing: (**a**) the initial collision, (**b**) the cloud of particles (CoP) formation and surface activation and (**c**) the surface contact with the moving contact point C, (**d**) dimensions of the 1-D model, (**e**) modeling scheme with the temporal course of the heat input and contact time.

**Table 4.** Material specific values (temperature independent during the simulations).

| Physical Quantity | Symbol | Unit | EN AW-6060 | C45 | Cu [31] |
|---|---|---|---|---|---|
| Density | $\rho$ | kg/m³ | 2700 [32] | 7700 [24] | 8960 |
| Heat capacity | c | J/kgK | 898 [32] | 470 [24] | 390 |
| Thermal conductivity | k | W/mK | 210 [32] | 42.6 [24] | 384 |
| Melting temperature | $T_{fus}$ | °C | 659 [31], pure aluminum | 1536 [31], pure iron | 1083 |
| Boiling temperature | $T_{vap}$ | °C | 2467 [31], pure aluminum | 3070 [31], pure iron | 2595 |
| Enthalpy of fusion | $\Delta H_{fus}$ | kJ/kg | 356 [31], pure aluminum | 276 [31], pure iron | 213 |

## 3. Results and Discussion

### 3.1. Effect of the Flyer Kinetics on the Material Flow

During the experimental study, the charging energy $E$ was increased stepwise until a circumferential weld seam was proved in the manual peel test. The minimum radial impact velocity for the Bmax setup was measured via photon Doppler velocimetry (PDV) and found to be approximately 340 m/s [17]. The corresponding cross sections of the 180° location are shown in Figure 5 and reveal small waves at the joining interface.

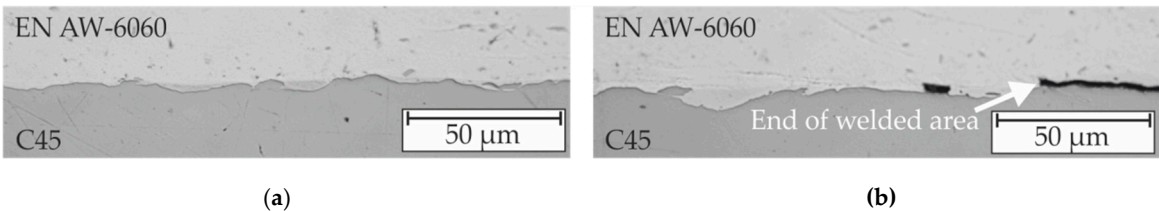

(**a**)                                                     (**b**)

**Figure 5.** Polished cross section of the MPW sample joined with the Bmax setup ($g$ = 1.5 mm, $E$ = 5.8 kJ, $I_{max}$ = 451 kA, Ø coil = 42 mm, $w_c$ = 10 mm, $l_w$ = 6 mm, welding direction from left to right, $v_{i,r} \approx$ 340 m/s measured with photon Doppler velocimetry (PDV) [17]): (**a**) in the middle and (**b**) at the end of the joining zone.

Although the required energy at the EmGen machine was higher compared to the Bmax setup, the minimum radial impact velocity was lower at approximately 250 m/s. One of the reasons is the difference in the flyer forming and collision behavior, which heavily depends on the discharging frequency [17]. The welding interface is almost smooth, see Figure 6.

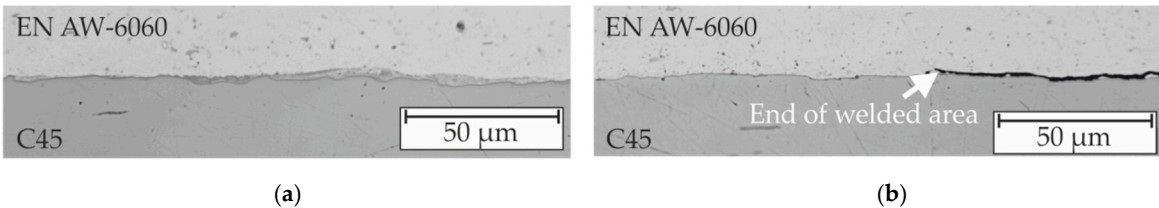

(**a**)                                                     (**b**)

**Figure 6.** Polished cross section of the MPW sample joined with the EmGen setup ($g$ = 1.5 mm, $E$ = 22.7 kJ, $I_{max}$ = 113 kA, Ø coil = 41 mm, $w_c$ = 15 mm, $l_w$ = 6 mm, welding direction from left to right, $v_{i,r} \approx$ 250 m/s calculated according to [22] with measured flash appearance time equal to 12.6 µs): (**a**) in the middle and (**b**) at the end of the joining zone.

The application of a 5 µm thick anodized layer on the flyer tube prevented the welding effect at the EmGen setup on the same energy level, but enabled the reconstruction of the material flow. Figure 7a shows the fragmented anodized layer that mainly stayed on its original position. The flyer material has been extruded through the interspaces. At the end of the joining zone, this effect was significantly reduced.

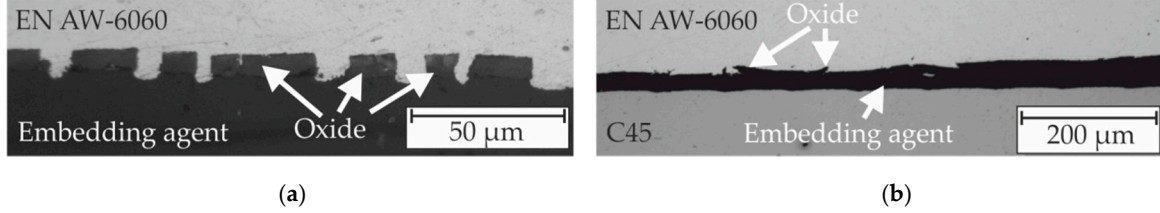

**Figure 7.** Polished cross section of the unwelded MPW sample joined with the EmGen setup (aluminum flyer anodized with 5 μm thickness, $g$ = 1.5 mm, $E$ = 22.7 kJ, $I_{max}$ = 117 kA, Ø coil = 41 mm, $w_c$ = 15 mm, $l_w$ = 6 mm, welding direction from left to right, $v_{i,r} \approx$ 250 m/s calculated according to [22] with measured flash appearance time equal to 12.6 μs): (**a**) in the middle (according to [17]) and (**b**) at the end of the joining zone.

If the radial impact velocity was increased to 460 m/s, the anodized layer in the middle of the joining zone was completely removed and a sound weld was generated, see Figure 8a. At the end of the joining zone, the anodized layer was embedded into the flyer material, which illustrates the massive material flow in Figure 8b. This phenomenon can occur at high speed impact situations and is called "jet". Jetting is associated with a high degree of plastic deformation and surface enlargement. It is also often described as a prerequisite for the surface cleaning and removing of oxides before the surfaces get in contact. The experiments described so far give clear evidence that this large amount of plastic deformation is not required for successful MPW, unless certain surface layers have to be removed or roughness peaks to be overcome. Furthermore, welding with velocities below the "jetting"-regime requires less impact velocity, which is beneficial for the lifetime of the tool coils due to the reduced mechanical and thermal loads. Thus, it is worthwhile to analyze the physical conditions that enable MPW at the lower process boundary.

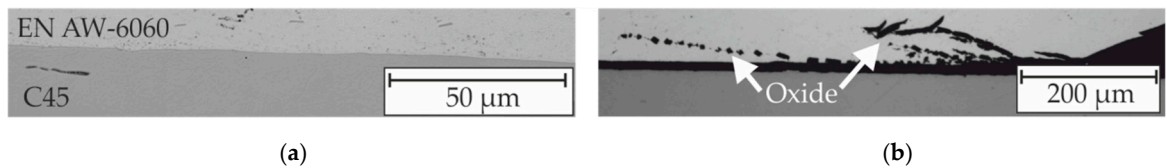

**Figure 8.** Polished cross section of the MPW sample joined with the Bmax setup (aluminum flyer anodized with 5 μm thickness, $g$ = 2 mm, $E$ = 9.6 kJ, $I_{max}$ = 533 kA, Ø coil = 42 mm, $w_c$ = 10 mm, $l_w$ = 6 mm, welding direction from left to right, $v_{i,r} \approx$ 460 m/s measured with PDV [33]): (**a**) in the middle (according to [33]) and (**b**) at the end of the joining zone.

During all successful MPW experiments a flash was visible. In a former study, it was shown that the high velocity impact flash occurred approximately 0.5 μs after the initial impact [22] and it was correlated with the weld seam formation [34]. Thus, the flash detection was utilized as a measurement system for the parameter adjustment and quality assurance during MPW [28,35]. The setup shown in Figure 2 was used to analyze the flyer kinetics and weld formation by increasing the parameter $d$ stepwise from zero to five millimeters. Thus, the initial collision point was shifted and the effect on the flash appearance time and weld formation was studied. In Figure 9, the average values of the flash appearance times are plotted for both MPW setups. The location of the slot of the coil or field shaper was not considered, since it differs significantly from the remaining circumference due to the reduced magnetic field intensity. The flash at the EmGen setup occurred more than 8 μs later. The measured flash appearance time of 19.5 μs corresponds to a radial impact velocity $v_{i,r}$ of approximately 160 m/s. Nevertheless, the gap closing time $\Delta t$ is much smaller for the EmGen setup, which seems to be an important factor for the weld establishment at these comparatively low radial impact velocities. At the EmGen setup, no flash was detected for $d$ = 5 mm and consequently, no weld seam was generated, see Figure 10. An experiment with $d$ = 10 mm, where the collision occurred between the flyer and the plastic spacer, did also not lead to an impact flash. This shows that the impact of the two metals itself

must be responsible for the flash initiation and not the compressed air within the closing gap between the flyer and the plastic spacer due to the high speed compression.

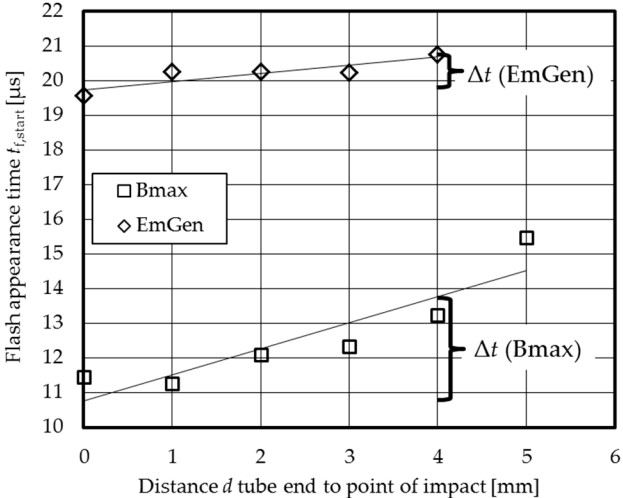

**Figure 9.** Mean values of flash appearance time $t_{f,start}$, referring to the rising tool coil current at $t = 0$ µs as depicted in the schematic oscilloscope in Figure 1, at 90°, 180°, and 270° for varied values $d$ for two MPW setups at the specific lower process boundary (Bmax: $g = 1.5$ mm, $E = 4.5$ kJ, $I_{max} = 380$ kA, Ø tool coil $= 41$ mm, $w_c = 10$ mm, $l_w = 6$ mm, EmGen: $g = 1.5$ mm, $E = 7$ kJ, $I_{max} = 70$ kA, Ø field shaper $= 41$ mm, $w_c = 10$ mm, $l_w = 6$ mm).

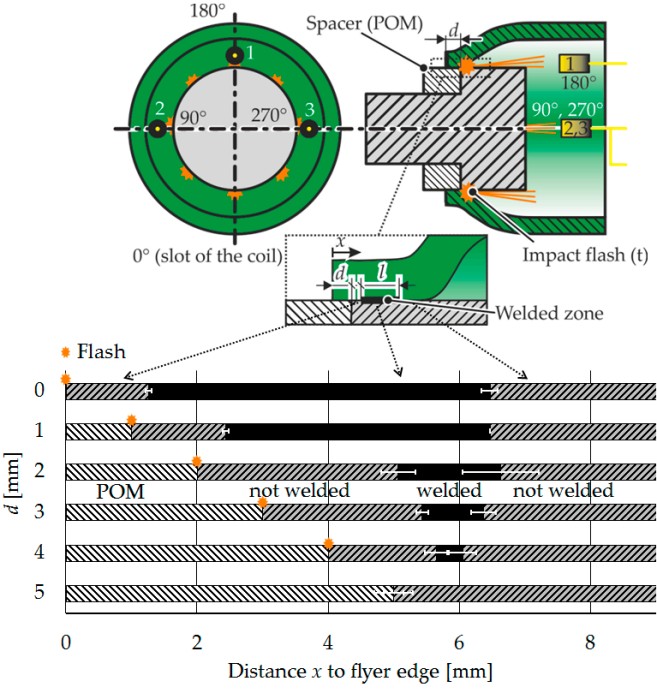

**Figure 10.** Mean values of welding start and end at 90°, 180°, and 270° for varied values $d$.

After the MPW experiment, the flyer was sheared from the parent part and the average weld seam length $l$ and its location were measured for each test, see Figure 10. The weld starts 1.2 mm after the first contact between flyer and parent, at the earliest. If the point of the initial metal impact was postponed by the parameter $d$, the weld formation was shifted, too. Thus, it seems reasonable that the high speed impact between the metallic partners and the corresponding impact flash are closely related to another necessary welding criterion, most likely the surface activation of the adjacent areas to be welded.

Basically, the necessity of the surface activation before welding is in good correlation with the "traditional" view of the role of the jetting effect and the corresponding surface enlargement. However, as shown in the first experiments, MPW is also possible at lower impact velocities where jetting and material flow are not initiated. To explain these findings, the mechanism presented by Deribas [6] and Ishutkin [5] seems to be a reasonable approach. They identified the appearance of a "cloud of particles" (CoP) at lower impact velocities. Obviously, the CoP plays an important role at the MPW process and, thus, it seems worthwhile to study this phenomenon in detail. The following section describes the investigation of the CoP with modified MPW experiments.

### 3.2. Characteristics of the "Cloud of Particles" (CoP)

The setup depicted in Figure 3 was used for a comprehensive investigation and characterization of the CoP. Therefore, the influence of the collision conditions on the appearance of the CoP and effect on the weld formation was studied. Furthermore, attendant effects like residues on the flyer and tempering colors on the parent surfaces outside the joining zone were recorded, see Figure 11.

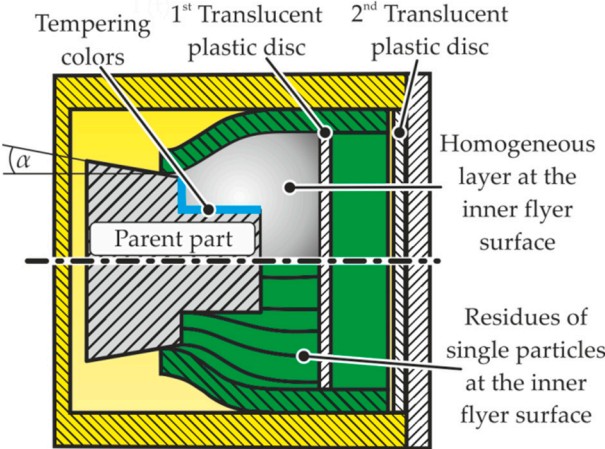

**Figure 11.** Effects on the flyer and parent surface (position of all parts is fixed during MPW).

A chamber was designed that enables MPW experiments on the Bmax-setup at different surrounding pressures. The maximum intensity of the process glare in vacuum is significantly reduced compared to the process in ambient atmosphere, see also [36]. Nevertheless, it is still visible, as shown in Figure 12.

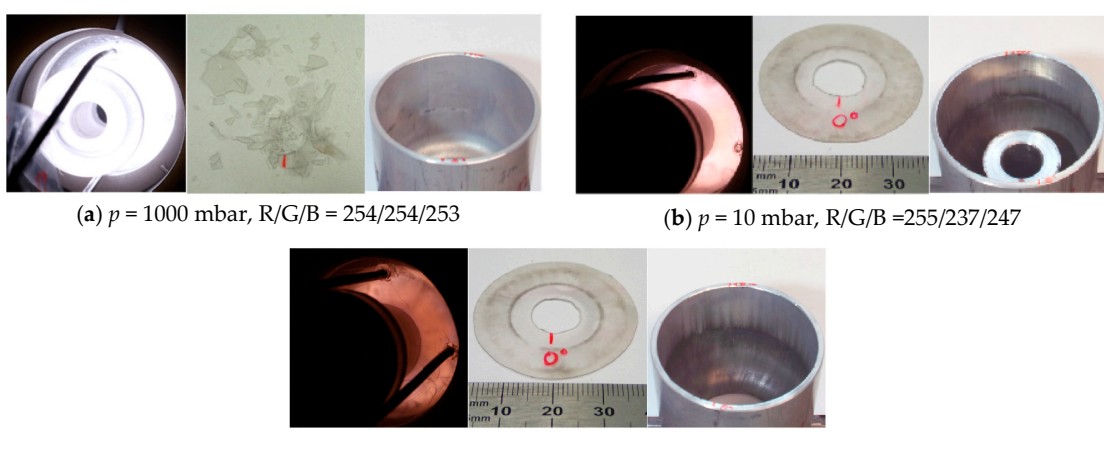

(**a**) $p$ = 1000 mbar, R/G/B = 254/254/253　　　　　(**b**) $p$ = 10 mbar, R/G/B =255/237/247

(**c**) $p$ = 0.1 mbar, R/G/B = 201/133/117

**Figure 12.** Long time exposures of the process glare, first plastic disc and flyer tube after MPW experiments at the Bmax setup ($l_c$ = 3 mm, $\alpha$ = 0°, $g$ = 1.5 mm, $E$ = 4.5 kJ, $I_{max}$ = 365 kA, Ø coil = 41 mm, $w_c$ = 10 mm, $l_w$ = 6 mm): (**a**) $p$ = 1000 mbar, (**b**) $p$ = 10 mbar, (**c**) $p$ = 0.1 mbar.

The rise time until the intensity reached its maximum at ambient atmosphere was 0.4 µs and it increased up to 3 µs in vacuum, see Figure 15. Since welding was possible in both cases, it can be concluded that a surrounding gas is not mandatory for welding. Probably, the CoP is formed at the initial impact, independent of the surrounding medium. The density of the medium in the joining gap determines the velocity of the jet [37] and it seems likely that shock compression of the surrounding gas occurs, which leads to the immediate light emission [16]. Furthermore, the shock compressed gas is the reason for the complete fracture of the first plastic disc during MPW in ambient atmosphere. In vacuum, there is no interaction with other gases and the CoP expands freely in the welding direction. At the same time, the residues at the inside of the flyer tube are noticeably increased, as well as the intensity of the tempering colors at the parent part. The starting time of the flash was almost identical for $p = 1000$ mbar ($t_{f,\text{start}} = 11.07$ µs) and $p = 0.1$ mbar ($t_{f,\text{start}} = 10.96$ µs). Thus, the flyer forming behavior is assumed to be independent of the gas pressure $p$.

The CoP formation was analyzed at $p = 0.1$ mbar by varying the collision length $l_c$ between 1 and 3 mm. Similar to [22], at ambient atmosphere, no flash was detectable at a collision length of 1 mm. No debris was deposited on the first plastic disc, as seen in Figure 13a. An increase of the collision length to 3 mm lead to a process glare, contaminants on the first plastic disc as depicted in Figure 13b and tempering colors on the parent part. Thus, the CoP was formed and heated up between 1 and 3 mm after the initial impact. The collision length $l_c$ was set to 3 mm for the following experiments.

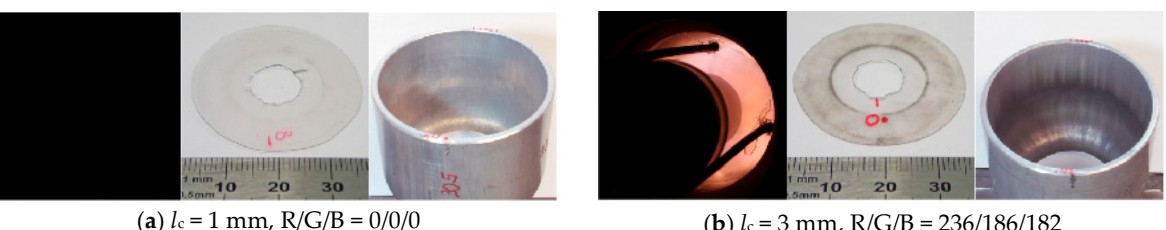

(a) $l_c = 1$ mm, R/G/B = 0/0/0        (b) $l_c = 3$ mm, R/G/B = 236/186/182

**Figure 13.** Long time exposures of the process glare, first plastic disc and flyer tube after MPW experiments at the Bmax setup ($p = 0.1$ mbar, $\alpha = 0°$, $g = 1.5$ mm, $E = 4.5$ kJ, $I_{\max} = 365$ kA, Ø coil = 41 mm, $w_c = 10$ mm, $l_w = 6$ mm): (a) $l_c = 1$ mm (b) $l_c = 3$ mm.

After investigating the CoP initiation at constant collision conditions, the appearance at different impact velocities and angles was studied systematically. Again, the process glare and attendant phenomena were recorded and compared, see Figure 14.

The collision angle was decreased by increasing the working length $l_w$. The intensity of the process glare increased significantly and finally its color changed from red to light blue for $l_w = 8$ mm. The first plastic disc fractured at the outer circumference at the largest collision angle at $l_w = 4$ mm and no weld was achieved. Furthermore, a circular crater was left at the second translucent plastic disc resulting from the harsh impact of single ejected particles that were guided by the inner flyer contour (see Figure 11). At a working length $l_w$ of 6 mm, a weld was generated. The marks at the first plastic disc and the inner flyer tube indicate a more vapor-like appearance of the CoP, since the residues are homogeneously distributed. This trend is continued for $l_w = 8$ mm. Here, the first plastic disc is almost clean, but the inside of the flyer tube is uniformly covered with a dark grey layer. Assuming an ideal black body emission, the R/G/B values enable the estimation of the CoP temperature. It is increased from 5500 K ($l_w = 6$ mm) to 8000 K ($l_w = 8$ mm), which is also reflected by the tempering colors of the parent parts. Due to the increased density of the CoP within the smaller volume of the joining gap, the heating, the fine dispersion of the CoP and, finally, the weld formation are supported. Similar effects were detected in the MPW experiment with increased charging energy $E = 6.5$ kJ, see Figure 14d. Here, the CoP was finely dispersed and on a high temperature level, too. The rise of the light intensity is shown in Figure 15 by the dotted line. The time interval $\Delta t$ between the formation of the CoP and the reaction with the first plastic disc is much shorter than in the reference experiment with $E = 4.5$ kJ. Thus, the average velocity of the CoP is higher and in the order of 10 km/s.

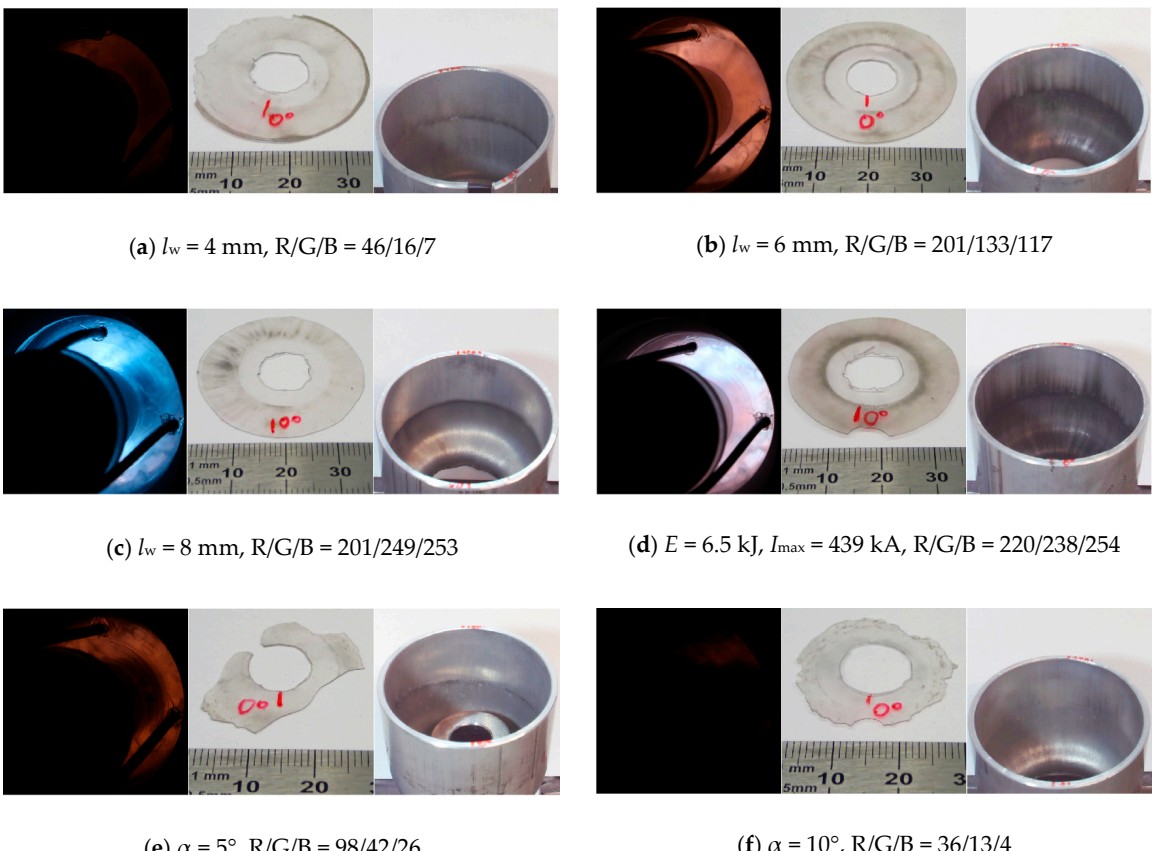

(**a**) $l_w$ = 4 mm, R/G/B = 46/16/7

(**b**) $l_w$ = 6 mm, R/G/B = 201/133/117

(**c**) $l_w$ = 8 mm, R/G/B = 201/249/253

(**d**) $E$ = 6.5 kJ, $I_{max}$ = 439 kA, R/G/B = 220/238/254

(**e**) $\alpha$ = 5°, R/G/B = 98/42/26

(**f**) $\alpha$ = 10°, R/G/B = 36/13/4

**Figure 14.** Long time exposures of the process glare, first plastic disc and flyer tube after MPW experiments at the Bmax setup (basic parameters: $l_c$ = 3 mm, $\alpha$ = 0°, $g$ = 1.5 mm, $E$ = 4.5 kJ, $I_{max}$ = 365 kA, Ø coil = 41 mm, $w_c$ = 10 mm, $l_w$ = 6 mm): (**a**–**c**) for different working length $l_w$, (**d**) charging energy, and (**e**,**f**) parent angles $\alpha$.

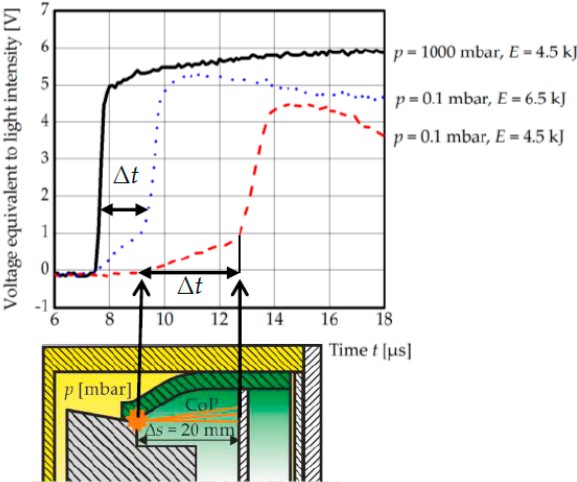

**Figure 15.** Light intensities for different surrounding pressures $p$ and charging energies at the Bmax setup ($t$ refers to the rising tool coil current as depicted in the schematic oscilloscope in Figure 1, $l_c$ = 3 mm, $\alpha$ = 0°, $g$ = 1.5 mm, Ø coil = 41 mm, $w_c$ = 10 mm, $l_w$ = 6 mm).

Chamfers at the parent part as depicted in Figure 3 lead to the destruction of the first plastic disc, a weaker process glare and a non-weld for $\alpha$ = 10° (see Figure 14e,f). In these cases, the degree of CoP compression and, thus, the temperature were lower.

### 3.3. Temperature Model

Although the temperature of the CoP was estimated in the previous chapter, the quantification of the heat transfer to the surfaces is still difficult since the density or mass of the CoP, as well as the surface coefficient for heat transfer, are hard to access. The knowledge of the heat input is essential for the following temperature model and, thus, a strategy based on the metallographic analysis of the welding result is applied. In a first step, the macroscopic distortion of the flyer tube and the parent part after MPW was analyzed as shown in Figure 16a. The initial wall thickness of 2 mm was reduced to 1.4 mm next to the initial impact zone of the free flyer edge and increased to 2.3 mm at the end of the close-fit zone. The indentation depth of the flyer into the parent was measured in analogy with [27] at different positions in the welding direction. The maximum value of 44 μm was found to be 0.5 mm behind the beginning of the welded zone and clearly below 10 μm at the end of the welded zone. At other MPW samples, the weld seams were even well established at positions without any indentation of the flyer into the parent material. Thus, plastic deformation might not be a necessary welding criterion, but is just a side effect of the high speed collision. In a second step and in order to study the thermal effects as another important welding criterion, the amount of melted flyer and parent material is quantified in polished cross sections. This value can vary along the welding direction, depending on the prevalent collision conditions as shown in Figure 16. The weld seam exhibits an almost smooth interface next to the start of the weld seam in Figure 16b and a characteristic wavy shape at the end of the welded zone in Figure 16c. The cross section of the pocket highlighted in Figure 16d is equal to a continuous melted layer with a thickness of approximately 6 μm. There are a few single iron particles visible in this pocket. As a consequence of the extreme high cooling rates that occur during welding, the volume in the pocket is "frozen" in a non-equilibrium state. Thus, a mixture of different phases can occur, including non-stoichiometric or metastable phases as reported by Bataev et al. [9] for different material combinations joined by EXW. Although the metallurgy was not studied in detail here, it is very likely that these types of phases occurred, too. The analysis of the chemical composition revealed an average ratio of 80 weight percent aluminum and 20 weight percent iron at the position of the line scans indicated in Figure 16d. Based on this ratio, the mass of molten aluminum $m_{Al}$ per area $A$ can be derived from Equations (1) to (6), where $b$ is the equivalent thickness of the molten layer, $m$ the mass, $V$ the volume, and ρ the density of aluminum, iron, or both elements, respectively. For this calculation, the material specific values in Table 4 are applied, while the index Al corresponds to EN AW-6060 and Fe to C45.

$$m_{Al}/m = m_{Al}/(m_{Al} + m_{Fe}) = 0.8 \tag{1}$$

$$m_{Fe} = (1/0.8 - 1) \times m_{Al} \tag{2}$$

$$V = V_{Al} + V_{Fe} = m_{Al}/\rho_{Al} + m_{Fe}/\rho_{Fe} = m_{Al}/\rho_{Al} + (1/0.8 - 1) \times m_{Al}/\rho_{Fe} \tag{3}$$

$$m_{Al} = V/(1/\rho_{Al} + (1/0.8 - 1)/\rho_{Fe}) \tag{4}$$

$$V = b \times A \tag{5}$$

$$m_{Al}/A = b/(1/\rho_{Al} + (1/0.8 - 1)/\rho_{Fe}) = 0.0149 \text{ kg/m}^2 \ (m_{Fe}/A = 0.00372 \text{ kg/m}^2) \tag{6}$$

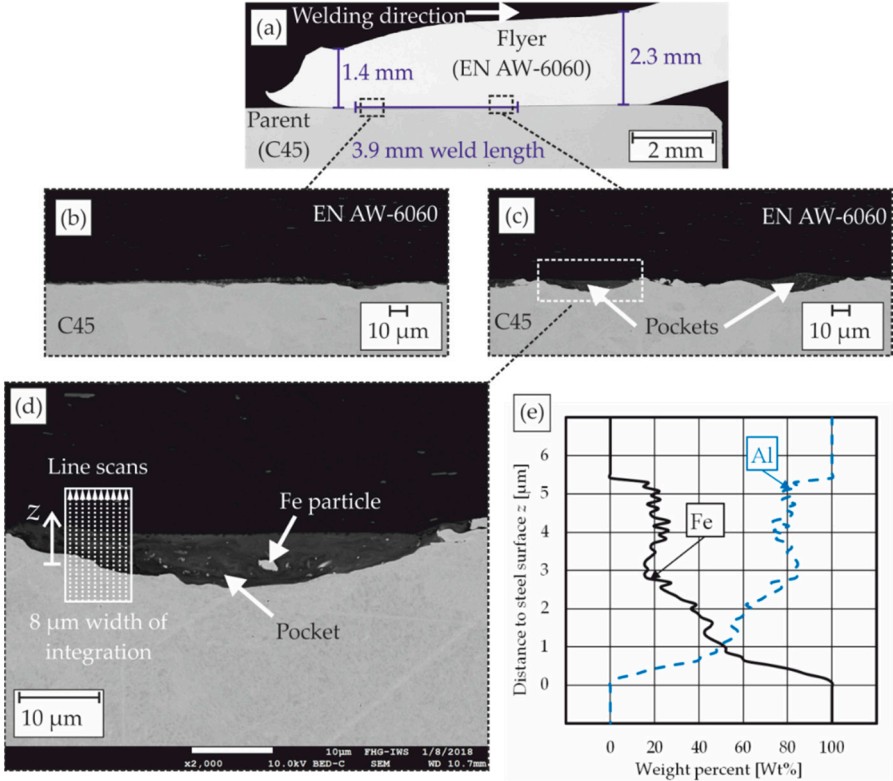

**Figure 16.** (**a**) Polished cross section of the joining zone after MPW at the Bmax setup ($l_w$ = 6 mm, $\alpha$ = 0°, $g$ = 1.5 mm, $E$ = 8.0 kJ, $I_{max}$ = 503 kA, Ø coil = 41 mm, $w_c$ = 10 mm, flyer tube thickness $s$ = 2 mm, flyer tube material in T66 condition), secondary electron image of the interface, (**b**) approximately 0.5 mm behind weld seam beginning with 500× magnification and approximately 3.5 mm behind weld seam beginning with (**c**) 500× magnification, (**d**) 2000× magnification and location of ten parallel line scans for EDS-analysis and (**e**) average element distribution perpendicular to the steel surface.

Now, the maximum heat input $Q_{Al}$ per area needed for the melting of aluminum can be calculated by applying Equations (7) [31] and (8) , taking the enthalpy of fusion $\Delta H_{fus}$ into account, and assuming a temperature-independent heat capacity $c$. Furthermore, the highest reachable temperature is the boiling temperature of the material, since possible metal vapor is assumed to be spewed out of the joining gap.

$$Q_{Al}/A = m_{Al}/A \times (c_{Al} \times \Delta T + \Delta H_{fus}) = 38.7 \text{ kJ/m}^2 \; (Q_{Fe}/A = 6.2 \text{ kJ/m}^2) \tag{7}$$

$$\Delta T \text{ [K]} = T_{vap} - 20\ ^\circ\text{C} \tag{8}$$

The total heat input $Q$ per area is now calculated according to (9) and is in good accordance with previous publications [15].

$$Q/A = Q_{Al}/A + Q_{Fe}/A = 44.9 \text{ kJ/m}^2 \tag{9}$$

To add plausibility to this calculated value, the kinetic energy of a flyer with a thickness $s$ of 1.5 mm and a radial impact velocity $v_{i,r}$ of 340 m/s according to Figure 5 by Equation (10) [31]. In this case, the total heat input is approximately one fifth of the kinetic flyer energy.

$$E_{kin}/A = s/2 \times \rho_{Al} \times (v_{i,r})^2 = 234.1 \text{ kJ/m}^2 \tag{10}$$

The total heat input calculated in (9) serves as an upper boundary in the one-dimensional thermodynamic COMSOL model and is assumed to be distributed equally to both surfaces. Thus, half

of the total heat input $Q$ is introduced into the aluminum and steel surfaces, respectively, and named as $Q_S$.

In the following section, the influence of certain input parameters for the numerical simulations is described. In each diagram, the results of a reference setup with the parameters given in Table 5, are plotted for a better comparison. Here, the maximum heat quantity is introduced into the surfaces within 0.5 µs, followed by an immediate contact of both joining partners.

**Table 5.** Parameters of the reference setup.

| Physical Quantity | Symbol | Unit | Value |
|---|---|---|---|
| Heat quantity at each surface | $Q_S/A$ | J/m$^2$ | 22,450 |
| Heating time | $t_{heat}$ | µs | 0.5 |
| Waiting time | $t_{wait}$ | µs | 0 |
| Flyer material | - | - | EN AW-6060 |
| Parent material | - | - | C45 |
| Consideration enthalpy of fusion? | - | - | true |

In the first simulation, the influence of the phase transitions on the surface temperatures was investigated, see Figure 17. The surface temperatures during the heating stage are almost identical, while they differ significantly during the cooling stage, exactly at the time when the melting point of aluminum is reached, approximately 1 µs after the contact $t_{con}$. The interface stays in the liquid phase for 0.6 µs, before the temperature decreases. This period might be critical for thin flyers, since the bounce-back effect occurs quite early.

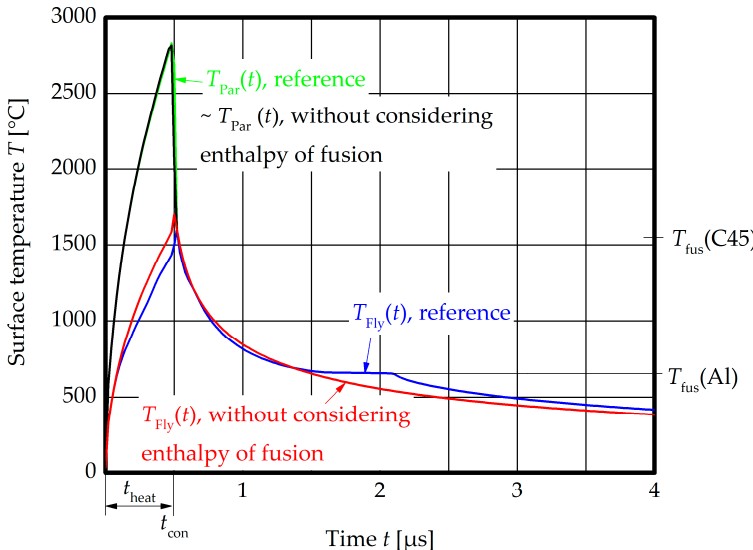

**Figure 17.** Comparison of $T_{Fly}(t)$ and $T_{Par}(t)$ for the reference setup (heat quantity at each surface 22,450 J/m$^2$, heating time 0.5 µs, waiting time 0 µs, flyer material EN AW-6060, parent material C45) with and without considering the enthalpy of fusion.

In Figure 18, the heat quantity $Q_S$ is reduced to 9100 J/m$^2$ at each surface, which was identified as a lower boundary in the analytical investigation of the melted volume in the interface. This value is insufficient to reach the melting temperature of steel and, thus, welding in the liquid phase is probably hindered.

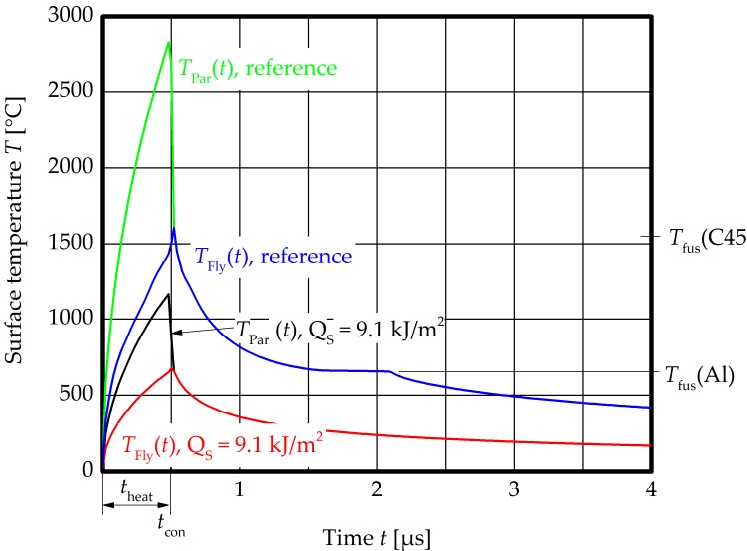

**Figure 18.** Comparison of $T_{Fly}(t)$ and $T_{Par}(t)$ for the reference setup and the setup with 9100 J/m$^2$ heat quantity at each surface.

Since the heating time $t_{heat}$ of the CoP is hard to assess experimentally, the influence was studied in the numerical simulation, see Figure 19. The increase of the heating time to 1 μs at a constant heat quantity leads to lower maximum temperatures after the heating period. Nevertheless, both materials are in the liquid phase at the time of contact $t_{con}$ and show a similar cooling behavior like the reference setup.

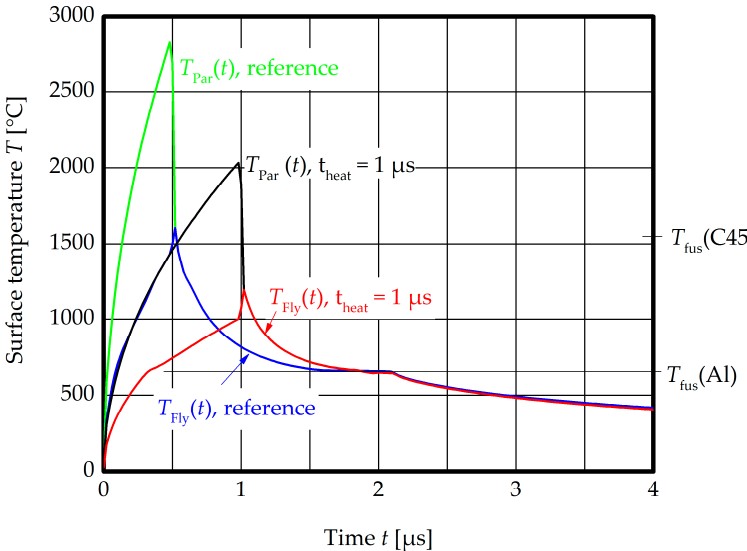

**Figure 19.** Comparison of $T_{Fly}(t)$ and $T_{Par}(t)$ for the reference setup and the setup with 1 μs heating time.

The waiting time $t_{wait}$ between the end of the heating and the contact $t_{con}$ was identified as an important factor. The cooling rate of the steel surface in the numerical simulation is in the order of $10^9$ K/s and, thus, it is solidified approximately 0.3 μs after the heating time. Figure 20 shows that a waiting time of 2 μs and even 1 μs until contact is too long for a liquid phase bonding, since the steel surface is already solidified at $t_{con1}$ and $t_{con2}$, respectively. The immediate contact after heating is necessary for this material combination, which corresponds to short gap closing times, high collision front velocities, or small collision angles, respectively. As explained in the previous chapter, this is

linked to an increased compression of the CoP and, thus, higher temperatures in the joining gap and heat input to both surfaces.

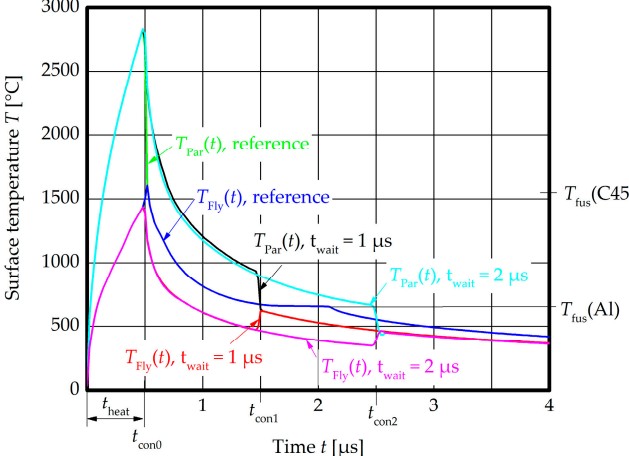

**Figure 20.** Comparison of $T_{\text{Fly}}(t)$ and $T_{\text{Par}}(t)$ for the reference setup and the setups with 1 µs and 2 µs waiting time.

The last set of simulations investigated the influence of the flyer material. The results for copper and C45 as flyer materials are plotted in Figure 21. From the theoretical point of view, liquid state bonding can be achieved in both cases. The surface of the copper reaches the melting temperature at the time of contact $t_{\text{con}}$ and enables a very fast cooling due to the high heat conductivity. The similar material combination C45 to C45 fulfills the requirements for liquid state bonding as well. Nevertheless, to establish a CoP with the same heat input like in the reference setup aluminum to steel, a higher impact velocity is needed due to the increased hardness of both joining partners. In such configurations, a soft interlayer material as described in [38] might be beneficial to reach a strong CoP at lower impact velocities.

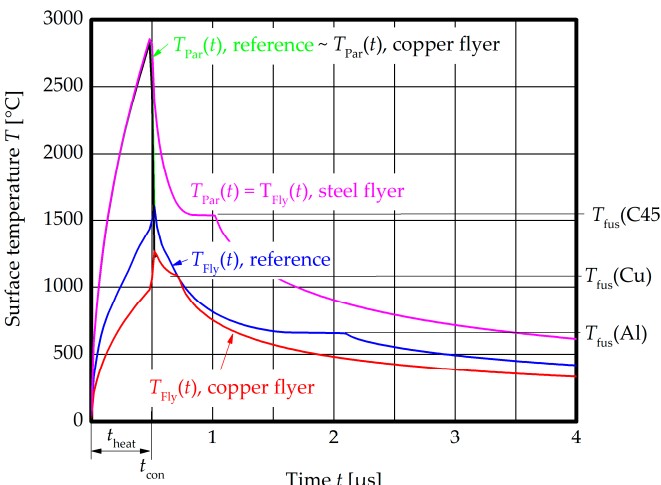

**Figure 21.** Comparison of $T_{\text{Fly}}(t)$ and $T_{\text{Par}}(t)$ for the reference setup and the setups with copper and C45 as flyer materials.

The results of the numerical simulations could be stated more precisely, if some of the input data would have been accessible during the experiments. Nevertheless, this numerical parameter study revealed the most important factors for MPW. An extension of the model will also include the reaction enthalpy for dissimilar metal welding, which was previously found to be beneficial for the weld formation at the lower process boundary [18]. Furthermore, the combination with mechanical

models that predict the time of bounce back effects will allow for a comprehensive prediction of the welding result.

## 4. Research Highlights

1. The experiments showed that jetting in the type of a strong material flow is not mandatory for a successful MPW process. A cloud of particles (CoP), which is ejected during the impact with lower velocities, enables welding, too. Compared to the "real" jet in the style of a massive material flow at higher impact velocities, the CoP cannot remove thick surface layers or facilitate welding with rough surfaces. In this case, an adapted surface preparation and cleaning process is essential to ensure a sufficient surface activation.

2. The appearance of the CoP and its effect on the weld formation is determined by the prevalent collision conditions, especially the collision angle. This factor can be adjusted by various machine related factors and the part geometries. Vacuum experiments show that the CoP is established during the first metal to metal impact with a certain minimum impact velocity. Afterwards, it is compressed in the closing joining gap, successively heated up and finally ejected in welding direction.

3. For small collision angles, the level of compression and the internal friction of the CoP are higher and, thus, the temperature in the joining gap increases. In this configuration, the CoP is finely distributed like a metal vapor, which activates the surfaces of the joining partners homogeneously and can be seen inside the flyer tube after the experiment. If the collision angle is increased, the temperature decreases and single macroscopic particles are ejected. These particles seem to have a reduced surface activation effect, compared to the finely dispersed metal vapor described previously and thus, inhibit welding.

4. These findings allow for an optimization of the energy input during MPW. If a small collision angle is ensured, the initial impact velocity can be reduced. Thus, less mechanical energy is required for the forming process and the loading on the tool coils is reduced with positive effects on their lifetime.

5. Normally, the MPW process is performed in ambient atmosphere, where the free CoP ejection is hindered by the surrounding air. This leads to a shock compression and sudden heat up of the gas and results in a very strong process glare. This strong lightning can be utilized for the quality assurance during industrial production [39].

6. The numerical simulations of the surface temperatures of both joining partners revealed a strong influence of the waiting time between the end of the heat input by the CoP and the contact of both joining partners. Especially for dissimilar metal welding, this time needs to be very low to avoid solidification before the contact. This finding is important for the theory of liquid state bonding and in good correlation with the experimental results. Small collision angles, or gap closing times, respectively, are beneficial for the weld formation during MPW.

## 5. Patents

The flash measurement system enables the parameter identification and quality assurance during production. It was patented for different impact welding processes [39,40].

**Author Contributions:** Conceptualization, J.B.; Funding acquisition, E.B. and A.E.T.; Investigation, J.B.; Methodology, J.B.; Project administration, E.B. and A.E.T.; Validation, J.L.-A., S.S., M.H., and S.G.; Writing—original draft, J.B.; Writing—review & editing, J.L.-A., S.S., M.H., and S.G.

**Funding:** This research was funded by the Deutsche Forschungsgemeinschaft (DFG, German Research Foundation), grant number BE 1875/30-3 and TE 508/39-3. This work is based on the results of subproject A1 of the priority program 1640 ("joining by plastic deformation"). We acknowledge support by the Open Access Publication Funds of the SLUB/TU Dresden.

**Acknowledgments:** We would like to acknowledge the effort for the sample preparation and SEM and EDS analysis at Fraunhofer IWS Dresden.

**Conflicts of Interest:** The authors declare no conflict of interest.

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
