# Peer review of "Thermal Effects in Dissimilar Magnetic Pulse Welding"

_metals, doi:10.3390/met9030348_

Round 1

Reviewer 1 Report

The article presents the thermal effects in dissimilar magnetic pulse welding. In the manuscript the effect of the fyler kinetics on the material flow, the characteristics od the "Cloud of Particles (CoP)" and the numerical simulations are presented.
In my opinion the experiment was designed appropriate, the methods were chosen properly. My overal merit of the work is very high.

I have some suggestions that could help to improve your work.

- I propose to add some references from Metal journal.

- In Introduction section, you can add real photo of your test stand.

- Table 3. - I propose to add the references in the name of the table, without 1, 2, 3. Now in table there is "3" like reference [24], and "3" in the Unit "kg/m3".

- Figure 5b. and Figure 6b. - please add the name of the areas (like in Fig 5a i 6a). Also please name the black area located near the boundry of the two areas.

- Please add references to your equations.

Author Response

Dear reviewer,

thanks for you helpful comments. According to your suggestions we included an additional reference from the journal “Metals”, a photograph of our test stand in Figure 3 and references to the most important equations. Furthermore we simplified the references in Table 3 (now Table 4) and named the areas in Figure 5 and 6.

Reviewer 2 Report

In this very interesting paper the main objective from this reviewer’s perspective is to better understand the conditions required to produce a sound weld between EN AW-6060 Aluminum alloy and C45 carbon steel, using the Magnetic Pulse Welding (MPW) process. The authors state more precisely that:

the objectives of this experimental and numerical work can be summarized as follows:

1.      Investigate the influence of the flyer kinetics on the material flow

2.      Study the influence of different collision conditions on the formation and properties of the jet or “cloud of particles” (CoP) and the corresponding thermal conditions in the joining gap

3.      Build up a temperature model for the welding interface based on the heat input by the CoP

A clever experimental setup provides a remarkable insight into the welding phenomena.

Micrographs of local surface melting are presented as a proof that the process is not only a cold welding process but involves also a heat source. The observation of a cloud of particles (CoP) ejected from the contact surface as the gap is closing, leads to the conclusion that the CoP generated at low contact angle is providing surface activation of the two parts in front of the contact line. With the proper CoP, a sound weld at lower energy (lower flyer kinetics) can be produced.

A numerical temperature model has been developed. The model assumes that the CoP provides the energy required to melt the surface of the aluminum flyer and the parent steel parts just before contact. The composition of the molten pockets observed in real welds is in the range of 80% Al - 20% Fe. Based on this model it is suggested that the waiting time between the end of the heat input by the CoP and the contact of both parts should be very short to allow a sound weld.  

Main comments:

-        Although the research presented is very good and interesting, it is not easy to read and understand this paper because there are three different setups and many parameters to be memorized while reading.  A nomenclature would help and each parameter should be described the first time they are used in the text or a figure.

-        There is not enough information in the paper to fully understand Figure 9 and Figure 15. What is t=0? It is mentioned that in a former study (ref 21) the impact flash occurred approximately 0.5 microseconds after the initial impact. The information is given in reference 21, where it is stated that “the rising current signal (of the generator) triggered the recording of the PDV signal and the light intensities”. The same information should be given in this paper.

-        The methodology used for building the numerical model does not seem right. Computing the heat input required to melt a thin layer (6 microns) of 80%Al-20% Fe does not provide the maximum heat input required to reach the melting temperature at the surface of the flyer or the parent parts because the heat losses by conduction in the parts are not taken into account. It can however be used as an initial approximation for the heat input in the COMSOL model. The second very important parameter is the heating time which has been fixed at 0.5 microseconds for the reference setup. According to figure 15, the duration of the impact flash is in the order of 10 microseconds, thus it can be deduced that the authors assume that the heat input generated by the CoP is transferred to the plates in the vicinity of the contact line. This assumption would be reasonable, but it is not clearly explained in the paper. What is the geometry and size of the moving zone receiving the heat input in the COMSOL model?

-        I don’t think that the very good experimental results presented in this paper are sufficient to prove that the CoP is the main contributor to the surface melting of the parts. Assumption 1 (line 177) does not seem reasonable.  The demonstration that the CoP has an effect on the surface activation of the parts is more convincing.

Author Response

Dear reviewer,

thanks for your helpful and detailed comments. According to your suggestion we included a table with the nomenclature used in the manuscript. Furthermore we added a description for the Trigger setup, the corresponding figure captions and updated the “display” of the oscilloscope in Figure 1 that shows the temporal course of the current in relation to the flash intensity. The COMSOL model takes the heat losses by conduction in the parts into account, although the diagrams in the manuscript just show the time-dependent temperature at the surfaces of the flyer and parent part. We modeled a depth of 100 µm as shown in Figure 4. Regarding the heating time and size of the moving zone we included some sentences in the description of the numerical model to explain the abstraction of kinematic process parameters like collision front velocity within our model. Last but not least we stated in assumption 1 that the CoP is responsible for the surface activation. Furthermore and in order to simplify the numerical model, it is assumed to be the only heat source (before the time of contact). The heat input by plastic deformation after the collision is not considered in the model.

Reviewer 3 Report

The subject is worthy and interesting, and it is one to which the authors can add significant contributions, but the paper needs minor changes. To make this paper publishable the authors need to expand data analysis, rewrite few sections of the results and discussion.

Here below my main comments:

>>376>> "the amount of melted flyer and parent material is quantified in polished cross sections. This value can vary along the welding direction, depending on the prevalent collision conditions as shown in Figure 16. The weld seam exhibits an almost smooth interface next to the start of the weld seam and a characteristic wavy shape at the end of the welded zone. The cross section of the pocket highlighted in Figure 16(d) is equal to a continuous melted layer with a thickness of approximately 6 μm. This volume consists of 80 weight percent aluminum and 20 weight percent iron".

The authors are strongly recommended to add detailed quantitative information on local chemical composition in different locations inside the "pocket" highlighted in Figure 16(d); Fe particle are clearly seen inside this "pocket". The authors are encouraged to discuss the metallurgy of the "pocket"; is the material a supersaturated solution of Fe in Al ? is it a metastable IMC ?

The authors should pay special attention to the distortions of the flyer tube and of the parent part. Detailed information should be added and discussed regarding the components' distortion and spring-back issues.

I hope above comments help to improve a future version of the paper.

Author Response

Dear reviewer,

thanks for your helpful comments. We were busy with the metallurgy in the joining interface. You are right, single Fe particles can be seen in the pocket – and are mentioned in the discussion, of course. The detailed analysis of the metallurgy of the “pocket” with grain sizes of a few nanometers would require other methods such as TEM (Goebel et al. 2012). Due to the extreme high cooling rates that occur during welding, the conventional phase diagram is not applicable. In this non-equilibrium, a mixture of different phases can occur, including non-stoichiometric or metastable phases. The last mentioned type was comprehensively studied by Bataev et al. (2017) for different material combinations. In our publication, we decided to confine our investigations to SEM and EDX-analysis. We think that this is sufficient to get the average element ratio and to estimate the energy input for the numerical model. We referred to the publication of Bataev et al., which presents the latest work in that field from our knowledge.

Furthermore, we performed an additional analysis to examine the distortion of both joining partners and included another section about this topic with an updated Fig. 16(a).